# Proof-of-concept for a non-invasive, portable, and wireless device for cardiovascular monitoring in pediatric patients

Jennifer C. Miller[1], Jennifer Shepherd[2,3], Derek Rinderknecht[4], Andrew L. Cheng[1,3], Niema M. Pahlevan[5,6]*

1 Division of Pediatric Cardiology, Children's Hospital Los Angeles, Los Angeles, CA, United States of America, 2 Fetal and Neonatal Institute, Division of Neonatology, Children's Hospital Los Angeles, Los Angeles, CA, United States of America, 3 Department of Pediatrics, Keck School of Medicine, University of Southern California, Los Angeles, CA, United States of America, 4 Chief Technical Officer Avicena, LLC, Pasadena, CA, United States of America, 5 Department of Aerospace and Mechanical Engineering, University of Southern California, Los Angeles, CA, United States of America, 6 Department of Medicine, Division of Cardiovascular Medicine, University of Southern California, Los Angeles, CA, United States of America

* pahlevan@usc.edu

**Data Availability Statement:** All relevant data are within the paper and its Supporting Information files.

## Abstract

Measurement of cardiac function is vital for the health of pediatric patients with heart disease. Standard tools to measure function including echocardiogram and magnetic residence imaging are time intensive, costly, and have limited accessibility. The Vivio is a novel, non-invasive, handheld device that screens for cardiac dysfunction by analyzing intrinsic frequencies (IF) $\omega_1$ and $\omega_2$ of carotid artery waveforms. Prior studies demonstrated that left ventricular ejection fraction can be derived from IFs in adults. This study 1) studies whether the Vivio can capture carotid arterial pulse waveform data in children ages 0–19 years old; 2) tests the performance of two sensor head geometries, one larger and smaller than the standard size used in adults, designed for the pediatric population; 3) compares the IFs between pediatric age groups and adults with normal function. The Vivio successfully measured a carotid artery waveform in all children over 5 years old and 28% of children under the age of five. The small head did not accurately measure a waveform in any age group. One-way analysis of variance (ANOVA) demonstrated a difference in the IF $\omega_1$ between the adult and pediatric cohorts (F = 7.3, Prob>F = 0.0001). Post host analysis demonstrated a difference between the adult cohort ($\omega_1$ = 99 +/- 5 bpm) and the cohorts ages 0–4 ($\omega_1$ = 111 +/- 2 bpm; p = 0.0006) and 15–19 years old ($\omega_1$ = 105 +/-5 bpm; p = 0.02). One-way ANOVA demonstrated a difference in the IF $\omega_2$ between the adult and pediatric cohorts (F = 4.8, Prob>F = 0.003), specifically between the adult ($\omega_2$ = 81 +/- 13 bpm) and age 0–4 cohorts ($\omega_2$ = 48 +/- 8 bpm; p = 0.002). These results suggest that the Vivio can be used to capture carotid pulse waveform data in pediatric populations and that the data produced can be used to measure intrinsic frequencies.

**Funding:** This study was partially funded by Atlantic Pediatric Device Consortium (APDC) Subaward No. RG219-G6 (http://atlanticpediatricdeviceconsortium.org/) to NP. The funder had no role in the study design, data collection and analysis, decision to publish, or preparation of the manuscript.

**Competing interests:** Niema M. Pahlevan hold equity in Avicena LLC and has consulting agreement with Avicena LLC. Derek Rinderknecht is the Chief Technical Officer of Avicena, LLC, the manufacturer of the Vivio and owns equity stake in the company. This does not alter our adherence to PLOS ONE policies on sharing data and materials.

## Introduction

Cardiovascular function is central to the health of pediatric patients with congenital and acquired heart disease, but the ability to accurately assess cardiac hemodynamics rapidly in a non-invasive manner is limited [1]. Echocardiography is currently the standard for non-invasive evaluation of left ventricular (LV) function in the pediatric population [2]. LV function is typically assessed by linear (M-mode) or two-dimensional echocardiographic indices such as the modified Simpson method or 5/6 area-length method [3–5]. Technical issues such as inadequate image quality and measurement variability based on assumptions that a single plane accurately represents a normally shaped ellipsoidal LV limit these methods [6–8]. Additionally, there is significant inter- and intra-observer variability in evaluation of LV function measures such as LV ejection fraction (LVEF) [9]. An echocardiogram requires time, specialized equipment, and health professionals trained in pediatric cardiology to perform and interpret the study, limiting assessment of ventricular function in pediatric patients to inpatient and outpatient centers specialized in pediatric cardiology. Other methods which are more accurate in measuring LV function such as 3D echocardiogram [10] and cardiac MRI [11] are even more limited in terms of accessibility, time intensiveness, and cost. A cardiac MRI can pose additional risks if anesthesia is required, as can occur in the case of infants and young children. Therefore, there remains a need for a non-invasive, inexpensive, and easy to use device for the diagnosis and monitoring of cardiovascular health in the pediatric population.

Pahlevan et. al [12] recently introduced a new mathematical method, called the intrinsic frequency (IF) method, that is suitable for the dynamic analysis of coupled physiological systems such as the LV and the arterial network. IF uses an integrative systems approach that considers the LV and arterial network as a coupled dynamical system (LV + arterial tree) that is decoupled upon closure of the aortic valve [13–16]. Previous clinical studies have shown that the IF method can be used to estimate LVEF from non-invasive arterial waveforms measured by an iPhone [17]. Recently, a wireless handheld prototype device (Vivio) capable of simultaneous collection of arterial pulse waveform and phonocardiogram data from the carotid artery was proven to accurately estimate LVEF using the IF method in a large study of adult patients with anthracycline exposure secondary to treatment of childhood cancer [18]. The Vivio requires little time as well as minimal training for use and was shown to measure LVEF more accurately than echocardiogram, with results comparable to the gold standard of cardiac MRI [18]. Given the range of neck sizes within the pediatric population, the Vivio was modified in this study to create both a smaller device head to fit more easily in small necks as well as a larger but more sensitive device to help reduce the amount of hand pressure required to collect a pulse signal.

This study explores whether the Vivio can be utilized in collecting carotid pressure waveforms and computing IFs in a pediatric population. The objectives of this study are to 1) evaluate whether the Vivio can be used to collect carotid artery waveforms required for IF calculations in patients ages 0 to 19 years; 2) compare the performance of two different sensor head geometries, one larger and one smaller, in a pediatric population to the standard sensor head size [18]; and 3) compare the range of intrinsic frequencies derived from the Vivio between different pediatric age groups and between children and adults with normal LV function.

## Materials and methods

### Study participants

A clinical study was conducted at Children's Hospital Los Angeles (CHLA). The Children's Hospital Los Angeles Institutional Review Board (IRB) approved the study protocol and

informed consent was obtained from the participants and/or legal guardians. Patients aged 0–19.9 years cared for at the CHLA Cardiology Clinic were invited to participate if an echocardiogram determined they had normal cardiac anatomy and ventricular function. This included post-cardiac transplant patients (n = 6) and post-chemotherapy patients with normal function on prior echocardiograms (n = 8). Exclusion criteria included patients with congenital heart disease or heart failure. A population of 28 adults ages 20–50 years with normal cardiac function and IFs collected from a prior IRB-approved study served as the comparison group [17].

## Standard cardiac evaluation: Echocardiogram

All echocardiographic studies were performed at CHLA using either an IE33 or Epiq 7 ultrasound system (Philips, Amsterdam, Netherlands). Images required for analysis included M-mode and 2D in the parasternal short axis at the mid-papillary muscle level of the LV, and 2D four-chamber view. Echocardiographic assessments were performed according to American Society of Echocardiogram guidelines. LV function was evaluated by measuring shortening fraction (FS) and ejection fraction (EF). FS was calculated from the LV end diastolic diameter (LVEDD) and LV end systolic diameter (LVESD) measurements in M-mode as: FS = 100*(LVEDD-LVEDS)/LVEDD. An IntelliSpace Cardiovascular Workstation (Philips), in apical 4-chamber view was used to calculate end-systolic and end-diastolic volumes using the modified Simpson's method. LV volumes were also measured via the 5/6 area-length method using the parasternal short axis view for area and apical four-chamber view for length. Using this method, LV volume was calculated at end-diastole and end-systole as: 5/6 * cross sectional area * length. LVEF percentage was calculated using the volumes derived from the modified Simpsons and 5/6 area-length method as: LVEF = 100*(LV end diastolic volume–LV end systolic volume)/(LV end diastolic volume) [4, 5]. LV volume calculations were performed by a single study investigator (J.M.).

## Wireless handheld device: Vivio

The Vivio operates like an optical tonometer that captures the carotid arterial waveform by recording the vibrations of the skin due to the pulse pressure wave as it propagates through the underlying vessel (e.g. carotid artery). The device uses a photoreflector that faces a reflective membrane which removes any variability in optical response to changes in skin tone. The Vivio was designed to address some of the shortcomings of conventional tonometry such as the requirement to have a rigid structure supporting the artery during measurement as well as probe perpendicularity. The incorporation of a membrane versus a probe tip affords the user greater freedom in the placement of the device over the pulse, which makes the carotid pulse measurement easier in pediatric patients. The Vivio system can be used with any mobile device or smartphone platform and therefore is inexpensive to deploy.

To better understand the effect of Vivio head geometry on the quality on carotid pulse waveforms collected in a pediatric population, three different head geometries were tested. The standard head size (inner diameter (ID) = 25 mm, outer diameter (OD) 30 mm) was the same as the adult version of the device used in previous studies [18]. Since the Vivio is a membrane-based tonometer there are two options to facilitate Vivio measurements in a pediatric population. One is to increase the size of the head, effectively lowering the membrane stiffness and reducing the pressure required to capture a carotid artery signal. The other is to decrease the geometry of the head to better accommodate the pediatric anatomy at the cost of increased pressure to capture a carotid artery signal due to increase membrane stiffness. For the smaller head, the ID was adjusted to 20 mm and the OD to 25 mm. For the larger head, the ID was

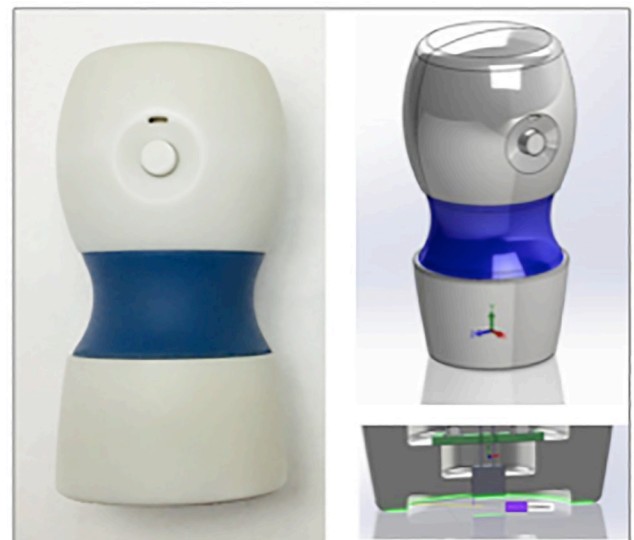

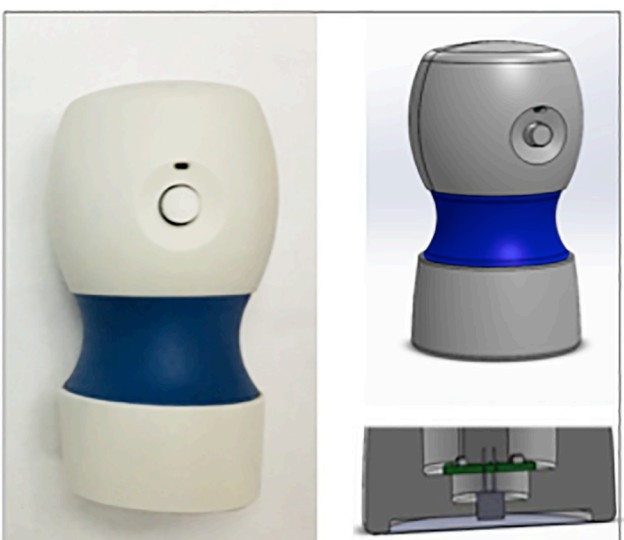

**Fig 1. Device design and Vivo head modifications.** Vivio device with an example and schematic of a large (A) and standard (B) head. (C). From left to right, comparison of the small (ID 20mm, OD 25 mm), standard (ID 25mm, OD 30 mm), and large heads (ID 30mm, OD 35mm). Abbr: Inner diameter (ID); outer diameter (OD).

adjusted to 30 mm and the OD to 35 mm. Fig 1 shows the device as well as the small, standard, and large heads. Adhesive contact thickness (difference in OD—ID = 5 mm) was kept the same in all heads.

The carotid pulse waveform of patients was recorded using the Vivio by gently holding the device against the neck of the subject over the carotid pulse for at least one minute. The data was transmitted via Bluetooth to an iPad Mini (Apple Inc., Cupertino, CA). The three different head sizes (small, standard, and large) were tested on each patient. A computer-generated algorithm selected at least three high quality cycles to produce an averaged cycle. One of the study investigators (N.P.) with expertise in the arterial hemodynamics and the IF method reviewed all waveforms to confirm the quality of the averaged cycle since the cycle detection

algorithm on the application looks for consistency in the pulse waveform data and therefore does not allow for sparse pulse waveform cycle capture. Since small artifacts present in low quality cycles such as tilted or distorted waveforms due to respiration or other motion cause large errors in the calculation of IFs, the investigator (N.P.) further reviewed signals not captured by the computer algorithm while blinded to the patient information and Vivio head size. A recording was considered acceptable if at least one cardiac cycle waveform of high quality could be selected for analysis. The selected cycles were used to compute IFs ($\omega_1$ and $\omega_2$). In brief, the IF algorithm computes the two intrinsic frequencies present within a cardiac cycle before and after the closure of the aortic valve, $\omega_1$ and $\omega_2$ respectively (see Intrinsic Frequency Method section below).

The three Vivio heads were compared in the age groups <1, 1–4 (1.0–4.99), 5–9 (5.0–9.99) and 10–14 (10.0–14.99) years of age to see if there was an advantage of using one head size over the other based on age. At least 5 patients were included in each age bracket. The operators spent at least 1 min for data collection. Evaluation of the carotid waveform was significantly limited by patient non-compliance under the age of 5 years old as well as anatomy constraints in the neonates and thus the age groups <1 and 1–4 years were combined for further analysis. Children 15–19 (15.0–19.99) years old were presumed to be adult-sized and not included in this comparison. The standard head size was used for these patients.

### Intrinsic frequency method

The mathematical formulation of the IF method is:

$$\text{Minimize}: \ \left\| p(t) - \chi(0, T_0)[(a_1\cos(\omega_1 t) + b_1\sin(\omega_1 t)] - \chi(T_0, T)[(a_2\cos(\omega_2 t) + b_2\sin(\omega_2 t)] - c \right\|_2^2,$$

This $L_2$ minimization is subject to continuity at $T_0$ and periodicity of the waveform. Here, $\chi(a, b)$ is the indicator function ($\chi(a,b) = 1$ if $a \leq t \leq b$ and $\chi(a,b) = 0$ otherwise), $p(t)$ is the carotid waveform, T is the period of the cardiac cycle, and $T_0$ is the left ventricle ejection time. In this study, we corrected $\omega_1$ and $\omega_2$ with LV ejection time and heart rate (HR) respectively. IF can be computed using this formula in a fraction of second. Details regarding mathematical formulation of IF method [12], its computational procedure [12], and its convergence/accuracy [19] can be found in previous publications [12, 17, 19].

### Statistical analysis

Characteristics of the study population were summarized using standard descriptive measures. $\omega_1$ and $\omega_2$ were compared between pediatric age groups 0–4, 5–9, 10–14, and 15–19 years and the adult cohort using a one-way ANOVA with post hoc testing using the Tukey test. Statistical calculations were performed in MATLAB 2018b (Mathworks, Natick, MA). A p-value < 0.05 was considered statistically significant.

## Results

Forty patients were enrolled with a median age of 6.7 years with a range of 0–19.4 years. This included 21 males (52.5%) and 19 females (47.5%). Eleven (27.5%) of these patients were ages <1 years, 7 (17.5%) were ages 1–4 years, 8 (20%) were ages 5–9 years, 8 (20%) were ages 10–14 years, and 6 (15%) were 15–19 years. Subjects weighed an average of 31.8 kg (2.7–107.9 kg) with an average BMI of 19.4 kg/m$^2$ (11.0–36.5 kg/m$^2$). All forty patients had structurally normal hearts with normal LV function. Of these patients, in the past, 6 (15%) had received a cardiac transplant and 8 (20%) had received chemotherapy. The complete demographics of the

**Table 1. Demographics.**

| Characteristics | Participants (N = 40) |
|---|---|
| **Sex, N (%)** | |
| **Male** | 21 (52.5%) |
| **Female** | 19 (47.5%) |
| **Race/Ethnicity, N (%)** | |
| **Asian/Pacific Islander** | 3 (7.5%) |
| **Black/African American** | 2 (5%) |
| **Hispanic** | 21 (52.5%) |
| **Non-Hispanic White** | 5 (12.5%) |
| **Other** | 9 (22.5%) |
| **Age at examination, years** | |
| **Median, range** | 6.7 (0.01–19.4) |
| **Age range, N (%)** | |
| **<1** | 11 (27.5%) |
| **1–4** | 7 (17.5%) |
| **5–9** | 8 (20%) |
| **10–14** | 8 (20%) |
| **15–19** | 6 (15%) |
| **Diagnosis, N (%)** | |
| **Normal** | 26 (65%) |
| **Post cardiac transplant** | 6 (15%) |
| **Post chemotherapy** | 8 (20%) |

patients enrolled are provided in Table 1. Further breakdown by age is illustrated in S1 Table in the supplementary material file. Hemodynamics per pediatric age cohort are shown in Table 2. As expected the average HR decreased while average blood pressure increased with age. Shortening fraction and ejection fraction measured by the modified Simpson's as well as 5/6 area-length methods showed no statistical difference across ages. All patients had normal function by these measures.

The three head sizes were tested in patients under the age of 15 years old. The small head size had weak and over-damped signals which did not allow for analysis of the carotid artery

**Table 2. Hemodynamics by pediatric cohort.**

| Age | <1 Year | 1–4 Years | 5–9 Years | 10–14 Years | 15–19 Years | |
|---|---|---|---|---|---|---|
| | Mean (Range, SD) | Mean (Range, SD) | Mean (Range, SD) | Mean (Range, SD) | Mean (Range, SD) | Significance |
| **Weight (kg)** | 5.2 (2.7–7.8, 1.5) | 13.1 (8.1–17.1, 2.7) | 32.8 (16–78.1, 21.4) | 56.1 (30.1–85.9, 16.2) | 61.3 (46.4–107.9, 22.5) | p<0.05 |
| **Height (cm)** | 57.5 (48–69.8, 6.5) | 91.4 (72–104, 10.4) | 123.3 (102.5–154, 18.3) | 153 (137–165.4, 9.9) | 164.2 (151–179.3, 10.1) | p<0.05 |
| **BMI (kg/m2)** | 15.4 (11–18.2, 2.3) | 15.6 (12.7–18.9, 2.0) | 19.6 (15.1–32.9, 6.5) | 23.8 (15–31.7, 5.9) | 25 (18.6–36.5, 6.3) | p<0.05 |
| **HR (bpm)** | 137 (104–158, 16) | 109 (88–125, 13) | 96 (81–123, 15.8) | 91 (71–125, 18.2) | 83 (68–95, 12.7) | p<0.05 |
| **SBP (mmHg)** | 91 (60–120, 17.8) | 98 (83–110, 11.9) | 106 (73–123, 15.3) | 114 (100–136, 11.1) | 119 (113–127, 5.5) | p<0.05 |
| **DBP (mmHg)** | 59 (40–91, 17.4) | 61 (50–76, 8.7) | 68 (55–77, 6.5) | 66.8 (55–96, 12.8) | 67 (58–81, 8.6) | p = 0.5 |
| **Echo SF (%)** | 38 (30.4–48.8, 5.5) | 36.8 (31.4–40.7, 3.5) | 40.8 (37.1–44.7, 2.8) | 37.5 (31.4–43.6, 4.3) | 37.8 (33.3–42.6, 4.1) | p = 0.43 |
| **Echo EF (%) \*** | 61.5 (57.4–65.4, 2.9) | 63.1 (57.6–74.5, 6) | 62.5 (56.9–71.4, 4.9) | 62.9 (55.7–69.5, 4.9) | 65.4 (58.5–72, 5.6) | p = 0.66 |
| **Echo EF (%)\*\*** | 61.9 (58.2–66.2, 2.9) | 63.6 (58.7–68.2, 2.9) | 62.4 (57.2–68.8, 4.8) | 63.7 (55.2–73.2, 5.9) | 65.4 (60–73, 4.7) | p = 0.58 |

\*Simpson's Method

\*\*5/6 Area-Length Method

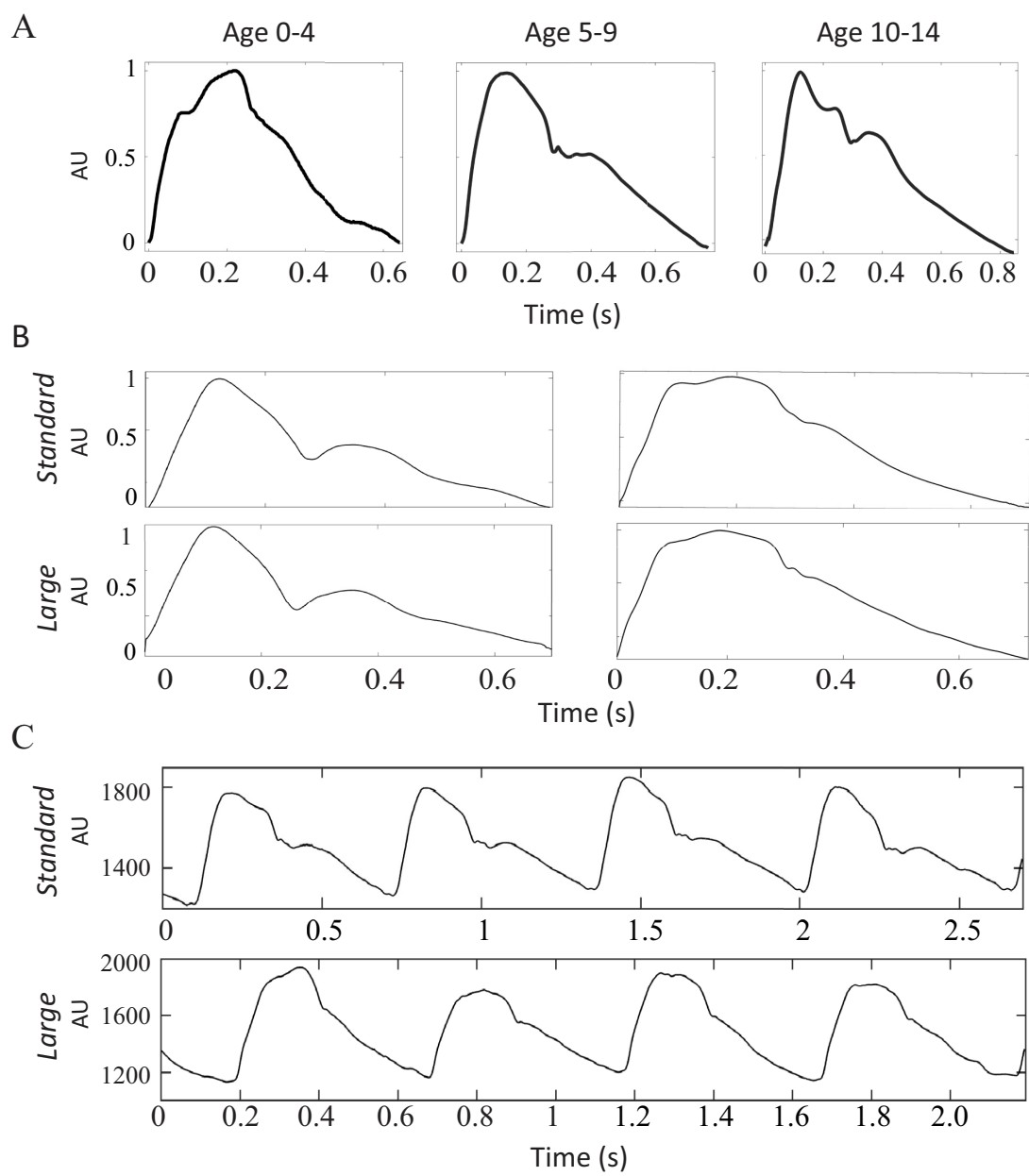

**Fig 2. Examples of carotid waveforms collected by Vivio.** (A) Examples of waveforms from three different age groups collected by Vivio (left: 0–4, middle: 5–9, right: 10–14). (B) Comparison of waveforms collected by both standard (upper row) and large (lower row) on the same patients (each column is a different patient). (C) Examples of multiple cycle tracings collected by a standard head (top row) and a large head (bottom row) in two different patients. AU is arbitrary unit.

waveform data. Both the standard and large heads demonstrated successful acquisition of carotid waveforms across all age ranges. Fig 2A displays examples of good quality waveforms from each pediatric age cohort (left: 0–4, middle: 5–9, right: 10–14). Fig 2B shows examples of averaged cycles from good quality recordings from two patients using both the standard (upper row) and large (lower row) heads. Fig 2C shows multiple cycle tracings of a good-quality recording in the two different patients using the standard (upper row) and large head (lower row). Comparison of the standard and large head sizes was made across age cohorts

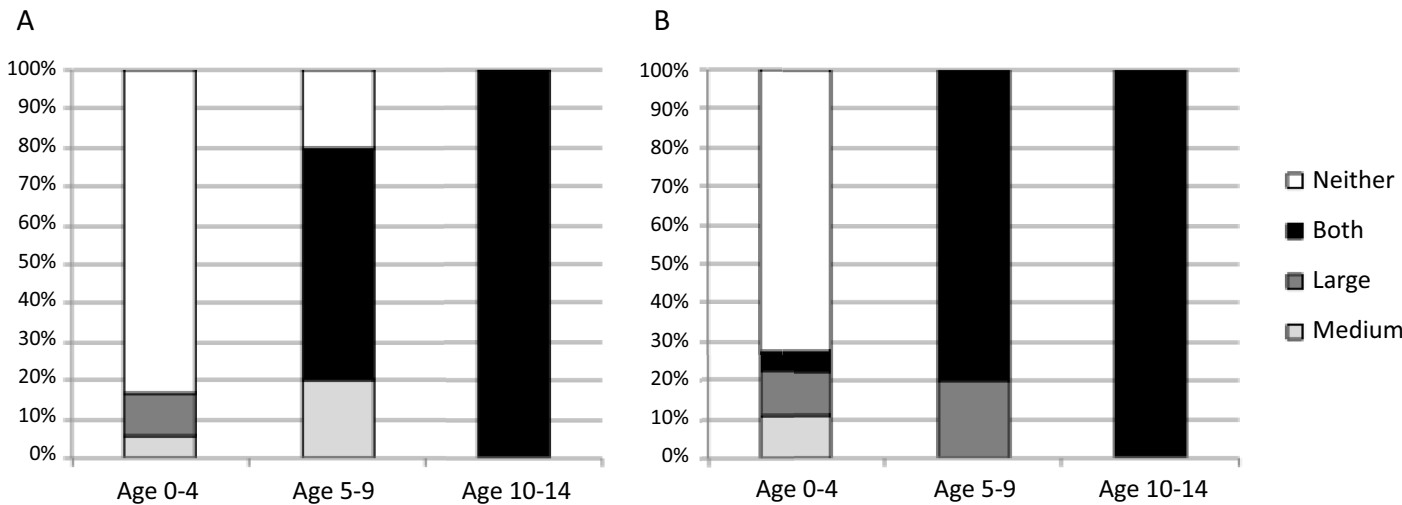

**Fig 3. Comparison of the standard and large Vivio heads.** Successful capture of carotid artery waveform as broken down by ages 0–4, 5–9, and 10–14 years as determined by (A) computer algorithm alone and (B) with additional manual review by IF expert.

0–4, 5–9, and 10–14 years of age. Fig 3 compares the success rate of capturing an analyzable waveform using the standard vs. large heads, by computer algorithm alone (Fig 3A) and with expert review (Fig 3B). Only 3 (17%) of the 18 patients under the age of 5 had analyzable waveforms. Those patients were still and positioned with their heads tilted slightly to the side and upwards to expose the carotid artery better. This improved to 5 patients (28%) after manual evaluation by the expert reviewer. The operator was directed to collect a signal for at least one minute. Among these patients, the standard head worked best in the neonates because the large head size was too large to fit under their chins. In the 5–9 year-old group, 4 (80%) of the 5 patients had signals that were analyzable by the computer algorithm. The fifth patient had an analyzable signal after evaluation by the expert reviewer. This patient had a tracheostomy collar, or breathing tube inserted through a surgically placed hole in the neck, and thus it is likely that the breath sounds from the tracheal collar affected data consistency. 100% of patients in the cohort age 10–14 had signals that were analyzable by the IF algorithm. The standard head was successful in collecting good quality signals in all patients while evaluation by the expert reviewer demonstrated successful signals in all patients with the large head as well. Based on this subgroup analysis, the large head data was used for patients < 10 years old and the standard head data was used for patients ≥10 years old for group comparisons of $\omega_1$ and $\omega_2$ when data were available from both head sizes.

The average $\omega_1$ was 105 +/- 5 bpm and the average $\omega_2$ was 69 +/- 14 bpm for the entire pediatric cohort. A one-way ANOVA comparison of $\omega_1$ across the pediatric and adult cohorts demonstrated a significant difference (F = 7.3, Prob>F = 0.0001). Post-hoc analysis revealed a significant difference between the adult cohort ($\omega_1$ = 99 +/-5 bpm) and both the 0–4 year old ($\omega_1$ = 111 +/- 2 bpm; p = 0.0006) and 15–19 year old ($\omega_1$ = 105 +/- 5 bpm; p = 0.02) cohorts (Fig 4A). A one-way ANOVA comparison of $\omega_2$ across all pediatric and adult cohorts showed a significant difference (F = 4.8, Prob>F = 0.003). Post hoc analysis identified the cohort age 0–4 ($\omega_2$ = 48 +/- 8 bpm) to be statistically different from the adult cohort ($\omega_2$ = 82+/- 13 bpm; p = 0.002) (Fig 4B). Fig 4C lists the $\omega_1$ and $\omega_2$ averages, range and standard error of the mean by age cohort.

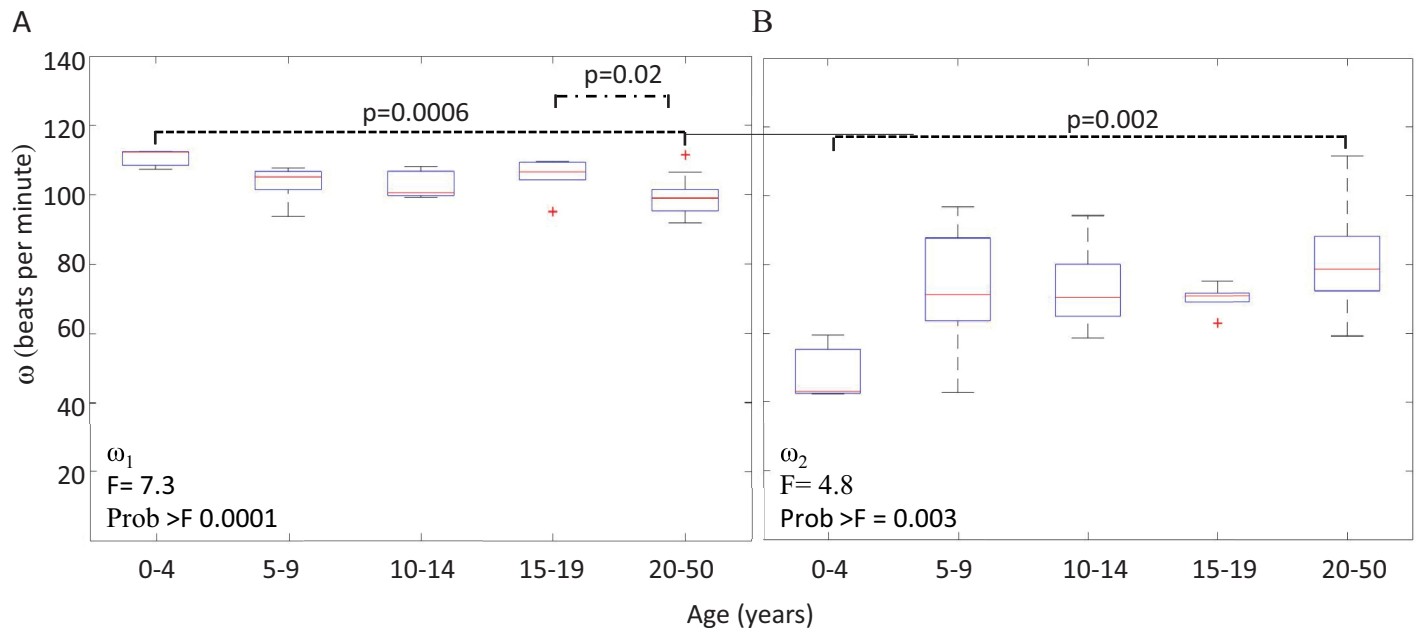

| | Age 0-4 | Age 5-9 | Age 10-14 | Age 15-19 | Age 0-19 | Adult 20-50 |
|---|---|---|---|---|---|---|
| | Mean (Range, SD) | Mean (Range, SD) | Mean (Range, SD) | Mean (Range, SD) | Mean (Range, SD) | Mean (Range, SD) |
| $\omega_1$ (bpm) | 111 (107-113, 2) | 103 (94-108, 5) | 103 (99-108, 4) | 105 (95-110,5) | 105 (94-113, 5) | 99 (92-111,5) |
| $\omega_2$ (bpm) | 48 (42-60, 8) | 73 (43-97, 17) | 73 (59-94, 11) | 70 (63-75,4) | 69 (42-97, 14) | 81 (59-112, 13) |

**Fig 4. Boxplot analysis of intrinsic frequencies amongst pediatric and adult populations.** One-way ANOVA was used to compare the intrinsic frequencies of (A) $\omega_1$ and (B) $\omega_2$ between ages 0–4, 5–9, 10–14, 15–19, and 20–50 years. (A) There was a significant difference in $\omega_1$ between the pediatric and adult cohorts (F = 7.3, Prob>F = 0.0001). Post-hoc analysis demonstrated a significant difference between the adult cohort to both the pediatric cohorts age 0–4 (p = 0.0006) and 15–19 (p = 0.02). (B) There was a significant difference in comparison of $\omega_2$ (F = 4.8, Prob>F = 0.003). Post hoc analysis demonstrated that a significant difference between the 0–4 years and age 20–50 years groups (p = 0.002). (C) $\omega_1$ and $\omega_2$ averages, range and standard error of the mean by age cohort.

## Discussion

This study demonstrates the feasibility of non-invasive measurement of carotid artery waveforms and evaluating IFs ($\omega_1$ and $\omega_2$) in children 5 years of age and older with normal cardiovascular anatomy and LV function using a handheld wireless device, the Vivio. We also showed the difference between IFs values ($\omega_1$ and $\omega_2$) of pediatric population and of adults with structurally normal hearts and normal LV function. This is clinically significant since physiologically relevant measures such as LVEF can be approximated from $\omega_1$ and $\omega_2$ as shown by Pahlevan et al [17] and Armenian et al [18]).

Given the wide range of body size in the pediatric population, the sensor head of the Vivio was modified to assess the ease-of-use and hand pressure required to capture non-invasive carotid artery waveforms in pediatric populations. Our results indicated that the small head was not sensitive enough to pick up adequate waveforms for evaluation in any age group. The standard and the large heads successfully captured analyzable waveforms in all age categories. In children under 10 the large head captured a good quality carotid artery signal more frequently than the standard head with the exception of infants aged 0–3 months (neonates). The slight advantage of the larger head in the smaller children is possibly derived from the ability of the large membrane to better amplify the vibratory signal. In neonates, the large head often covered the entire neck leading to significant noise and motion artifact from breathing and oral motion. Additionally, the device hand-grip base length made it difficult to position the head flush on the neck secondary to short neck anatomy without significant repositioning of

the neonate which resulted in irritation. In the future, the utility of the Vivio in pediatric population may be enhanced by reducing the length of the body of the Vivio device to better fit the neck space of neonates.

The Vivo successfully captured the carotid artery waveform in all children greater than 5 years of age. The Vivio best captured the carotid artery waveform when the subject held his/her neck in a position to expose the carotid triangle. This was achieved by rotating the head laterally 30–60 degrees and tilting the head up about 30 degrees. Neonates have a characteristically shorter neck in part secondary to a larger head to body ratio and lack of muscle development and thus require a more significant superior tilt of the chin up to 60 degrees to expose the carotid triangle. The Vivio was successful in capturing the carotid artery waveform in children less than 5 years who were able to hold still with their necks in this position without making noise as motion and sound create significant artifact. Methods to achieve this position included placing a neck roll beneath the back in the case of neonates, positioning the child in the parents lap sideways and leaning the head on the parents shoulder, and having the child sit in the parents lap with a video of interest positioned slightly up and to the side to achieve the angle of the neck desired. The device itself was less irritating to younger children than the process of finding the carotid pulse, and so it is possible the success rate may have significantly improved if the operator spent more time (e.g. 5 min) measuring for the waveform after giving the child time to settle. Furthermore, younger children are often fearful of strangers as well as unknown devices, particularly in unfamiliar settings, leading to significant motion artifact. It is likely that with parental training this device can be used by parents with whom the patients are more comfortable or in the setting of sleep at home. Parents were amenable to using the device themselves based on the simple application of the device.

The IF algorithm was used to extrapolate IFs $\omega_1$ and $\omega_2$ from the carotid artery waveforms collected by the Vivio. Previous studies have computed IFs among adults with the goal of estimating LVEF using IF $\omega_1$ and $\omega_2$ [17, 18]. Since it is difficult to obtain MRI cardiac function data in children (secondary to both cost and possible need for anesthesia in this population), IF evaluation by Vivio holds significant promise for a quick and easy method of accurately measuring LVEF in pediatrics. This study shows no statistical difference between pediatric cohorts for IF $\omega_1$ and $\omega_2$; however, there was a statistical difference between cohort age 0–4 years and the adult population in both $\omega_1$ and $\omega_2$. There was also a statistical difference in $\omega_1$ between the adult population and cohort age 15–19 years. These differences may, particularly in the cohort age 0–4 years, be secondary to the low number of successfully captured waveforms. It is important to note that removal of the cohort age 0–4 may change the statistical significance of $\omega_1$ and $\omega_2$ particularly with larger cohort sizes; however, given the great need for a better method to measure LVEF in this population these data were included in analysis. It will be important to reanalyze this age group in the future after modifications are made to allow for effective use of the device in this age group. A limitation of this study overall is the low number of patients enrolled and so future studies with larger patient enrollment will be important to verify the lack of difference seen between the pediatric cohorts in addition to the difference seen in specific pediatric cohorts in comparison to the adult population. The equation that computes LVEF from IFs in adults is a function of $\omega_1$ and also depends on $\omega_2$ [17, 18]. This study suggests that the normal range of $\omega_1$ and $\omega_2$ among pediatrics is different from adults. This is expected since cardiovascular dynamics in pediatrics is different than in adults (e.g. different HRs and different blood distribution to upper/lower body). This study was limited to patients with normal LV function. Further studies will be required to enroll patients with various stages of heart failure and a wide range of LVEF in order deduce an equation to calculate LVEF using IFs in the pediatric population similar to the study performed by Pahlevan *et al* [17].

## Conclusion

This study demonstrates the proof-of-concept that the Vivio can be used to successfully measure the carotid artery waveform for evaluation of cardiovascular intrinsic frequencies (IFs) in children 5 years of age and older. Our results showed a statistically significant difference in IF $\omega_1$ and $\omega_2$ between children and adult populations suggesting that further studies are required to derive an equation relating IFs to LVEF in the pediatric population. Parental training for data collection or creation of a hands-free version of the Vivio could significantly improve the utility of the Vivio in pediatric patients especially among patients under the age of 5 years. Overall, this study suggests that Vivio (a non-invasive, portable, and smartphone-based device) has the potential for cardiovascular monitoring in pediatric patients in an at home environment.

## Supporting information

**S1 Table. Breakdown of physiology by pediatric cohort.** Number of patients with each type of physiology by age group.
(TIFF)

## Author Contributions

**Conceptualization:** Derek Rinderknecht, Andrew L. Cheng, Niema M. Pahlevan.

**Data curation:** Jennifer C. Miller, Niema M. Pahlevan.

**Formal analysis:** Jennifer C. Miller, Niema M. Pahlevan.

**Funding acquisition:** Niema M. Pahlevan.

**Investigation:** Jennifer C. Miller, Jennifer Shepherd, Andrew L. Cheng, Niema M. Pahlevan.

**Methodology:** Jennifer C. Miller, Andrew L. Cheng, Niema M. Pahlevan.

**Project administration:** Andrew L. Cheng, Niema M. Pahlevan.

**Resources:** Niema M. Pahlevan.

**Software:** Derek Rinderknecht, Niema M. Pahlevan.

**Supervision:** Andrew L. Cheng, Niema M. Pahlevan.

**Validation:** Andrew L. Cheng, Niema M. Pahlevan.

**Visualization:** Niema M. Pahlevan.

**Writing – original draft:** Jennifer C. Miller, Niema M. Pahlevan.

**Writing – review & editing:** Jennifer C. Miller, Jennifer Shepherd, Derek Rinderknecht, Andrew L. Cheng, Niema M. Pahlevan.

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
