## [Decision Letter · Decision Letter 0]

10 Oct 2019

PONE-D-19-22751

Proof-Of-Concept For A Non-invasive, Portable, and Wireless Device the Vivio for Cardiovascular Monitoring in Pediatric Patient

PLOS ONE

Dear Miller,

Thank you for submitting your manuscript to PLOS ONE. After careful consideration, we feel that it has merit but does not fully meet PLOS ONE’s publication criteria as it currently stands. Therefore, we invite you to submit a revised version of the manuscript that addresses the points raised during the review process.

Please address the comments below, specifically commenting on feasibility concerns. , 

We would appreciate receiving your revised manuscript by Nov 24 2019 11:59PM. To enhance the reproducibility of your results, we recommend that if applicable you deposit your laboratory protocols in protocols.io, where a protocol can be assigned its own identifier (DOI) such that it can be cited independently in the future. For instructions see: http://journals.plos.org/plosone/s/submission-guidelines#loc-laboratory-protocols

We look forward to receiving your revised manuscript.

Best regards,

John Lynn Jefferies, MD MPH FACC FAHA

Academic Editor

PLOS ONE

**Journal Requirements:**

2.  Thank you for including your ethics statement:  "This study was approved by the CHLA IRB CHLA-17-00377. Consent/assent was obtained verbally form the participant and/or their parent as appropriate.".

a.Please amend your current ethics statement to include the full name of the ethics committee/institutional review board(s) that approved your specific study.

b.Once you have amended this/these statement(s) in the Methods section of the manuscript, please add the same text to the “Ethics Statement” field of the submission form (via “Edit Submission”).

"N.P.: This study was partially funded by Atlantic Pediatric Device Consortium (APDC) Subaward No. RG219-G6 http://atlanticpediatricdeviceconsortium.org/

The funders had no role in the study design, data collection and analysis, decision to publish, or preparation of the manuscript."

"Niema M. Pahlevan hold equity in Avicena LLC  and has consulting agreement with Avicena LLC.

Derek Rinderknecht is the Chief Technical Officer of Avicena, LLC, the manufacturer of the Vivio and owns equity stake in the company."

**Comments to the Author**

1. Is the manuscript technically sound, and do the data support the conclusions?

Reviewer #1: Yes

Reviewer #2: Partly

2. Has the statistical analysis been performed appropriately and rigorously? 

Reviewer #1: Yes

Reviewer #2: Yes

3. Have the authors made all data underlying the findings in their manuscript fully available?

Reviewer #1: Yes

Reviewer #2: Yes

4. Is the manuscript presented in an intelligible fashion and written in standard English?

Reviewer #1: Yes

Reviewer #2: Yes

5. Review Comments to the Author

Reviewer #1: The concept, study design, analysis, and interpretation seem appropriate. There are no ethical concerns. A further discussion of the challenges with the device in younger children would be helpful to the reader.

Reviewer #2: This article reports on the feasibility of the use of a non-invasive, portable cardiovascular monitoring system, the Vivio in pediatric patients.

METHODS: For the age groupings do the authors actually mean 1-4, 5-9, 10-15, 16-20? There seemed to be overlap in the age groups 1-5, 5-10, 10-15, 15-20 in some places and >15 in others, 16-20. However, both probes appear to have been used in the 15 and under, but only the standard adult probe in the 16 and older group. Please clarify, otherwise it appears that some ages are double counted and “groupings” seem off.

RESULTS: Despite finding statistically significant differences between children and adults the numbers in each pediatric group are quite small, especially in the less than 5 age group, in which only 3-5 patients had an analyzable wave forms. I am not sure the frequency data (IF) from the Vivio should even be reported for the less than 5 age group, with larger numbers statistical significance could actually be lost.

Page 12, line 237: The part of the sentence that states: the success rate may have significantly improved….this should be deleted from results and only be in the discussion section.

DISCUSSION: On page 14 line 281 add: in children older than 5 years, the feasibility is not really proven for the younger set.

On page 15 lines 303-305: This is new information about the data being capturable in those less than 5 who held still or had their heads tilted. This should somehow be incorporated into the results when the data collection on this age group is reported. (I take it these were the 3-5 who had interpretable data)

On page 16 lines 314-315: Lack of differences between the pediatric cohorts for IF’s could also be due to numbers.

On page 16, lines 321- 323: This study “suggests” not “demonstrates,” as it seems the differences were due to the youngest and oldest nonadult age group which seems odd.

6. PLOS authors have the option to publish the peer review history of their article (what does this mean?). If published, this will include your full peer review and any attached files.

Reviewer #1: No

Reviewer #2: No

---

## [Author Response · Author response to Decision Letter 0]

20 Nov 2019

We thank our reviewers for the positive evaluations and constructive suggestions. Our manuscript has been greatly improved with the reviewers’ help. Below we list our responses in the sequence in which they were raised in each reviewer’s report. These responses are included in the Response to Reviewers document as well.

Comments to the Author

Reviewer 1:

Comment 1: A further discussion of the challenges with the device in younger children would be helpful to the reader.

Response:

Thank you for your review and the thoughtful comments. We have revised the Discussion in the manuscript as follows to further discuss the challenges of the device in younger children.

“The Vivio successfully captured the carotid artery waveform in all children greater than 5 years of age. The Vivio best captured the carotid artery waveform when the subject held his/her neck in a position to expose the carotid triangle. This was achieved by rotating the head laterally 30-60 degrees and tilting the head up about 30 degrees. Neonates have a characteristically shorter neck in part secondary to a larger head to body ratio and lack of muscle development, and thus require a more significant superior tilt of the chin up to 60 degrees to expose the carotid triangle. The Vivio was successful in capturing the carotid artery waveform in children less than 5 years who were able to hold still with their necks in this position without making noise, as motion and sound create significant artifact. Methods to achieve this position included placing a neck roll beneath the back in the case of neonates, positioning the child in the parents lap sideways and leaning the head on the parent’s shoulder, and having the child sit in the parent’s lap with a video of interest positioned slightly up and to the side to achieve the angle of the neck desired. The device itself was less irritating to younger children than the process of finding the carotid pulse, and so it is possible the success rate may have significantly improved if the operator spent more time (e.g. 5 min) measuring for the waveform after giving the child time to settle. Furthermore, younger children are often fearful of strangers as well as unknown devices, particularly in unfamiliar settings, leading to significant motion artifact. It is likely that with parental training this device can be used by parents with whom the patients are more comfortable or in the setting of sleep at home. Parents were amenable to using the device themselves based on the simple application of the device.”

Reviewer 2: 

Comment 1a: METHODS: For the age groupings do the authors actually mean 1-4, 5-9, 10-15, 16-20? There seemed to be overlap in the age groups 1-5, 5-10, 10-15, 15-20 in some places and >15 in others, 16-20. 

Response: 

We would like to thank the reviewer for this constructive comment. We have changed the group labels to <1, 1-4, 5-9, 10-14, and 15-19. These adjustments were made throughout in the document, tables, and figures as necessary. For further clarification, in the Methods we stated that we enrolled patients 0-19.9 years old and additionally defined age groups as follows so there would be no confusion about age cut offs:

“The three Vivio heads were compared in the age groups <1, 1-4 (1.0-4.99), 5-9 (5.0-9.99) and 10-14 (10.0-14.99) years of age to see if there was an advantage of using one head size over the other based on age.” . . . . “Children 15 – 19 (15.0-19.99) years old were presumed to be adult-sized and not included in this comparison. The standard head size was used for these patients”

In the Results section we included:

“Forty patients were enrolled with a median age of 6.7 years with a range of 0-19.4 years.”

Comment 1b: However, both probes appear to have been used in the 15 and under, but only the standard adult probe in the 16 and older group. Please clarify, otherwise it appears that some ages are double counted and “groupings” seem off. 

Response: 

 In the original analysis, we included one patient age 15.4 years old in the comparison of the 3 head sizes. For this reason, we stated we only used the standard probe in patients 16 and older. Removing this patient does not alter our overall conclusion that there was successful capture of analyzable waveform in all patients age 10-14. We thus removed this patient and recreated the Fig 3 in order to keep consistent age cohorts throughout the paper to minimize confusion. Removal of the 15.4 year-old patient resulted in all patients with computer analyzable signals using both medium and large heads. This adjustment was made to Fig 3a. There were no changes to Fig 3b. 

In making this edit, we noted that Fig 3 was created based on the original and not final computer averaged data set. The final computer averaged data set used for analysis in the rest of the paper was able to measure a waveform using both the standard and large head in a patient previously only noted have a wave form measured in the large head in the age group 5-9 years. This adjustment was made in Fig 3a. There was no difference in Fig 3b or to the conclusion described in the paper that only 80% of patients had signals that were analyzable by computer algorithm alone.

Comment 2: RESULTS: Despite finding statistically significant differences between children and adults the numbers in each pediatric group are quite small, especially in the less than 5 age group, in which only 3-5 patients had an analyzable wave forms. I am not sure the frequency data (IF) from the Vivio should even be reported for the less than 5 age group, with larger numbers statistical significance could actually be lost. 

Response: 

We would like to thank the reviewer for this constructive comment. When the less than 5 years of age group is removed, there is a significant difference in ω1 between the pediatric and adults cohorts (prob>F 0.0027). Post-hoc analysis demonstrates the difference is between the adult and age 15-19 cohort (p=0.003). Thus, removal of the age cohort 0-4 does not change the conclusion in regards to ω1 and we will need to further data in the future to deduce an equation for LVEF. Removal of the age group less than 5 years of age shows that there is not a significant difference in ω2 between the adult and pediatric cohorts (prob>F 0.16). This is in line with the subgroup analysis provided in the paper that demonstrates a significant difference only between the age cohorts 0-4 and the adult population. We agree that keeping this population in our analysis has the possibility to alter the results of ω2; however, given that there is still a difference in ω1 and the fact that this population in particular is in need of a better method for measuring LVEF, we feel it is important to include the age cohort in our analysis. We have modified our discussion to address the reviewers concern as follows: 

“It is important to note that removal of the cohort age 0-4 may change the statistical significance of ω1 and ω2 particularly with larger cohort sizes; however, given the great need for a better method to measure LVEF in this population these data were included in analysis” 

Comment 3: Page 12, line 237: The part of the sentence that states: the success rate may have significantly improved….this should be deleted from results and only be in the discussion section. 

Response: 

Thank you for the recommended revision. We revised the manuscript as recommended. The sentence was deleted from the results and added to discussion.

“The device itself was less irritating to younger children then the process of finding the carotid pulse, and so it is possible the success rate may have significantly improved if the operator spent more time (e.g. 5 min) measuring for the waveform after giving the child time to settle”

Comment 4: On page 14 line 281 add: in children older than 5 years, the feasibility is not really proven for the younger set. 

Response: 

Thank you for the recommended revision. We have revised the manuscript to include the phrase “in children 5 years of age and older” 

Comment 5: On page 15 lines 303-305: This is new information about the data being capturable in those less than 5 who held still or had their heads tilted. This should somehow be incorporated into the results when the data collection on this age group is reported. (I take it these were the 3-5 who had interpretable data) 

Response: 

Thank you for the recommended revision. We have revised the results section of the manuscript to include the sentence “Those patients were still and positioned with their heads tilted slightly to the side and upwards to expose the carotid artery better.” 

Comment 6: On page 16 lines 314-315: Lack of differences between the pediatric cohorts for IF’s could also be due to numbers. 

Response: 

Thank you for the constructive comment. We agree with the reviewer’s comment that this is an important limitation. We have revised the discussion section of the manuscript to include the following statements in order to address the comments provided: 

“A limitation of this study overall is the low number of patients enrolled and so future studies with larger patient enrollment will be important to verify the lack of difference seen between the pediatric cohorts in addition to the difference seen in specific pediatric cohorts in comparison to the adult population.” 

Comment 7: On page 16, lines 321- 323: This study “suggests” not “demonstrates,” as it seems the differences were due to the youngest and oldest nonadult age group which seems odd.

Response: 

We have revised the manuscript as requested. 

Additional response to editor: Table 2 was not introduced in the text of the body. We added the following sentence to the results section to address this issue:

“Hemodynamics per pediatric age cohort are shown in Table 2.”

---

## [Decision Letter · Decision Letter 1]

4 Dec 2019

PONE-D-19-22751R1

Proof-Of-Concept For A Non-invasive, Portable, and Wireless Device for Cardiovascular Monitoring in Pediatric Patients

PLOS ONE

Dear Dr. Miller,

Thank you for submitting your manuscript to PLOS ONE. After careful consideration, we feel that it has merit but does not fully meet PLOS ONE’s publication criteria as it currently stands. Therefore, we invite you to submit a revised version of the manuscript that addresses the points raised during the review process.

We would appreciate receiving your revised manuscript by Jan 18 2020 11:59PM. To enhance the reproducibility of your results, we recommend that if applicable you deposit your laboratory protocols in protocols.io, where a protocol can be assigned its own identifier (DOI) such that it can be cited independently in the future. For instructions see: http://journals.plos.org/plosone/s/submission-guidelines#loc-laboratory-protocols

We look forward to receiving your revised manuscript.

Kind regards,

John Lynn Jefferies, MD, MPH, FACC, FAHA

Academic Editor

PLOS ONE

Reviewers' comments:

Reviewer's Responses to Questions

**Comments to the Author**

1. If the authors have adequately addressed your comments raised in a previous round of review and you feel that this manuscript is now acceptable for publication, you may indicate that here to bypass the “Comments to the Author” section, enter your conflict of interest statement in the “Confidential to Editor” section, and submit your "Accept" recommendation.

Reviewer #1: All comments have been addressed

Reviewer #2: (No Response)

2. Is the manuscript technically sound, and do the data support the conclusions?

Reviewer #1: Yes

Reviewer #2: Partly

3. Has the statistical analysis been performed appropriately and rigorously? 

Reviewer #1: Yes

Reviewer #2: I Don't Know

4. Have the authors made all data underlying the findings in their manuscript fully available?

Reviewer #1: Yes

Reviewer #2: Yes

5. Is the manuscript presented in an intelligible fashion and written in standard English?

Reviewer #1: Yes

Reviewer #2: Yes

6. Review Comments to the Author

Reviewer #1: The authors responded adequately to the reviewers' comments. The study provides important new insights.

Reviewer #2: The issues that I had previously brought up have been addressed within the limitations of the study.

I have a couple of new comments.

Under study participants post transplant patients n=5 and post chemotherapy patients n=9. However in the results section post transpalnt n=6 and post chemo n=8 in the text and Table 1. Please reconcile.

Page 13 line 260: I think the authors mean B not A when referring to figure 3. "A" is used twice.

The authors report that after manual review some additional waveforms could be added for assessment that the computer algorithm misclassified as not usuable. Were there any reversed by manual review, meaning the computer algorithm said the waveforms were usable but the manual reviewer felt they actually were not usable?

7. PLOS authors have the option to publish the peer review history of their article (what does this mean?). If published, this will include your full peer review and any attached files.

Reviewer #1: No

Reviewer #2: No

---

## [Author Response · Author response to Decision Letter 1]

9 Dec 2019

We thank our editor and reviewers for the positive evaluations and constructive suggestions. Our manuscript has been greatly improved with the reviewers’ help. Below we list our responses in the sequence in which they were raised in each reviewer’s report. 

Comments to the Author

Reviewer #1: The authors responded adequately to the reviewers' comments. The study provides important new insights.

Response:

We thank the author for their response

Reviewer #2: 

Comment 1: The issues that I had previously brought up have been addressed within the limitations of the study. I have a couple of new comments.

Under study participants post transplant patients n=5 and post chemotherapy patients n=9. However in the results section post transpalnt n=6 and post chemo n=8 in the text and Table 1. Please reconcile.

Response: 

We would like to thank the reviewer for this constructive comment. We have revised the section study participants to match the results section. Sup Fig 1 was also reconciled. The table and results discussion were reviewed and are correct.

Comment 2: Page 13 line 260: I think the authors mean B not A when referring to figure 3. "A" is used twice.

Response: 

We thank the reviewer for their careful review. We revised the manuscript as recommended and changed the Fig 3 legend to list A and B.

Comment 3: The authors report that after manual review some additional waveforms could be added for assessment that the computer algorithm misclassified as not usuable. Were there any reversed by manual review, meaning the computer algorithm 

Response: 

We would like to thank the reviewer for this question. The automatic computer algorithm looks at the consistency of the recordings. For example, if there are 1-2 appropriate waveforms interspaced with poor recordings the algorithm will not capture them; however, a waveform expert can easily identify the good cycles in this situation. The reverse is highly unlikely. No computer identified recordings were later deemed by the expert reviewer to be inaccurate in this paper.

---

## [Editor Report · Decision Letter 2]

13 Dec 2019

Proof-Of-Concept For A Non-invasive, Portable, and Wireless Device for Cardiovascular Monitoring in Pediatric Patients

PONE-D-19-22751R2

Dear Dr. Miller,

We are pleased to inform you that your manuscript has been judged scientifically suitable for publication and will be formally accepted for publication once it complies with all outstanding technical requirements.

Best regards,

John Lynn Jefferies, MD MPH FACC FAHA

Academic Editor

PLOS ONE
---

## [Editor Report · Acceptance letter]

20 Dec 2019

PONE-D-19-22751R2 

Proof-Of-Concept For A Non-invasive, Portable, and Wireless Device for Cardiovascular Monitoring in Pediatric Patients 

Dear Dr. Miller:

I am pleased to inform you that your manuscript has been deemed suitable for publication in PLOS ONE. Congratulations! Your manuscript is now with our production department. 

With kind regards,

on behalf of

Dr. John Lynn Jefferies 

Academic Editor

PLOS ONE